# Stress Management Intervention for Leaders Increases Nighttime SDANN: Results from a Randomized Controlled Trial

**DOI:** 10.3390/ijerph19073841

**Published:** 2022-03-24

**Authors:** Elisabeth Maria Balint, Peter Angerer, Harald Guendel, Birgitt Marten-Mittag, Marc N. Jarczok

**Affiliations:** 1Department of Psychosomatic Medicine and Psychotherapy, Ulm University Medical Center, 89081 Ulm, Germany; elisabeth.balint.research@gmail.com (E.M.B.); marc.jarczok@uni-ulm.de (M.N.J.); 2Burnout Section, Privatklinik Meiringen, 3860 Meiringen, Switzerland; 3Institute of Occupational, Social and Environmental Medicine, Centre for Health and Society, Faculty of Medicine, Heinrich Heine University, 40204 Düsseldorf, Germany; peter.angerer@hhu.de; 4Department of Psychosomatic Medicine and Psychotherapy, Klinikum rechts der Isar, Technische Universität Muenchen, 81675 Munich, Germany; b.marten-mittag@tum.de

**Keywords:** heart rate variability, work stress, occupational health, intervention, RCT, physiological outcome measures

## Abstract

Stress management interventions aim to reduce the disease risk that is heightened by work stress. Possible pathways of risk reduction include improvements in the autonomous nervous system, which is indexed by the measurement of heart rate variability (HRV). A randomized controlled trial on improving stress management skills at work was conducted to investigate the effects of intervention on HRV. A total of 174 lower management employees were randomized into either the waiting list control group (CG) or the intervention group (IG) receiving a 2-day stress management training program and another half-day booster after four and six months. In the trial, 24 h HRV was measured at baseline and after 12 months. Heart rate (HR), root mean square of successive differences (RMSSD), standard deviation of normal-to-normal intervals (SDNN), and standard deviation of the average of normal-to-normal intervals (SDANN) were calculated for 24 h and nighttime periods. Age-adjusted multilevel mixed effects linear regressions with unstructured covariance, time as a random coefficient, and time × group interaction with the according likelihood-ratio tests were calculated. The linear mixed-effect regression models showed neither group effects between IG and CG at baseline nor time effects between baseline and follow-up for SDANN (24 h), SDNN (24 h and nighttime), RMSSD (24 h and nighttime), and HR (24 h and nighttime). Nighttime SDANN significantly improved in the intervention group (z = 2.04, *p* = 0.041) compared to the control group. The objective stress axis measures (SDANN) showed successful stress reduction due to the training. Nighttime SDANN was strongly associated with minimum HR. Though the effects were small and only visible at night, it is highly remarkable that 3 days of intervention achieved a measurable effect considering that stress is only one of many factors that can influence HR and HRV.

## 1. Introduction

Workplaces can bring stability and safety and add to a person’s health when working conditions are adequate [1]. However, when work is a source of chronic stress, health deteriorates and the risk of stress-associated diseases, such as coronary heart disease [2,3], and common mental disorders (CMDs), such as depression, increases [4,5,6]. The risk of incident coronary heart disease and stroke is increased 1.1- to 1.6-fold in individuals with stress in their work or private life [7]. In men with cardiometabolic disease, job stress contributes to mortality independently of conventional risk factors and their treatment [8]. Chronic stress is linked to increased systemic inflammation and coagulation [9], thereby contributing to plaque disruption and thrombus formation. Additionally, it is correlated with behavioral risk factors, such as smoking [10]. Therefore, European guidelines recognize stress as a risk factor for cardiovascular disease and recommend interventions to improve stress management [11].

The underlying biological pathways that explain these associations include the two primary stress axes and the central autonomic network (CAN), which coordinates them. The two axes are the hypothalamus–pituitary–adrenal (HPA) axis [12] and the autonomic nervous system (ANS) [13,14]. The latter can be indexed by heart rate variability (HRV). The range of the variation of heart rate mirrors the capacity of the body to adapt to environmental challenges [15,16], such as work demands and emotion regulation [13]. To achieve this, it needs an activating part, the sympathetic nervous system (SNS), and a part to restore the resources, which is represented by the parasympathetic nervous system (PNS). The most prominent nerve of the PNS is the Vagus nerve, which is a fast and bidirectional route between the brain and periphery [17,18]. HRV has also been shown to correlate with physical [19,20] and mental [21] health and predicts mortality [22]. It shows a circadian rhythm, with vagal activity peaking at nighttime [23,24]. Blunted circadian rhythms with reduced vagal activity at nighttime are associated with adverse health outcomes, namely metabolic syndrome [25] and increased stroke risk [26]. To reflect the circadian rhythm, a 24 h measurement is necessary and daytime and nighttime should be investigated separately, as mean HRV may not reflect a disturbed rhythm [24].

The association between high work stressor demands, as measured by different stress models including the effort-reward imbalance (ERI) model or the job demand-control model, and reduced HRV is strong [27,28,29]. Thus, HRV may reflect the physiological correlates of an organism’s strain described by these models that precedes manifest disease.

The prevention of disease by reducing stress in the workplace is an established research area [30]. One approach is the use of stress management interventions (SMIs) in the workplace [31,32]. However, evidence for their efficacy in terms of physiological measures is scarce and refers mainly to SMIs that are specifically designed for patients with cardiovascular disease. Here, a reduction in physiological risk has been shown in terms of reduced blood pressure and heart rate after psychosocial skills training [33]. For interventions conducted in the workplace with a primary prevention focus, most studies rely on the self-report measures of stress to test the effect of their intervention [31,32]. Though self-report measures, such as self-rated health, do correlate with disease [34], physiological measures would be desirable. Of course, a direct measure of morbidity and mortality reduction is difficult to obtain as the period between intervention and disease onset can be decades in a healthy working population; meanwhile, other influencing factors usually occur as well. A practical physiological measure serving as a proxy for the physiological changes induced by the investigated intervention is necessary. HRV not only represents a physiological parameter correlated with both stress and disease, but it also has properties that facilitate wider application in that the required RR interval measurement is non-invasive and can be obtained at low cost. With advances in digital health technologies, techniques using deep learning, artificial intelligence, and neural networks are emerging [35,36,37,38], some of which can approximate heart rate without the need for a device to be worn directly on the skin [39,40]. This rapidly developing area holds the potential for widespread application, especially for the field of preventive medicine in the workplace.

Within the working population, managerial positions were particularly associated with high strain [41,42]. Middle and lower management are particularly stressed groups within companies. They find themselves in a sandwich position since on the one hand, they have to assume responsibility for the employees under their control and on the other hand, they have to implement guidelines from higher management without being able to influence them. Therefore, an SMI was designed for this target group based on the ERI model [27]. The aims of the SMI were to improve the awareness of potential stressors in the workplace, improve the ability to cope with stressful situations, empower employees to influence workplace conditions in the long term, and enhance social support in the workplace [43]. The 12-month on-site randomized controlled trial investigated the effects of the SMI on subjective and physiological stress markers. Limm et al. [43] have already reported the long-term effects of this SMI on the self-reported stress reactivity scale and salivary alpha-amylase, which both improved significantly.

Although the relationship between work stress and HRV has been investigated by several studies [44], very few studies have covered the effects of SMIs on HRV. Wetzel et al. investigated a small group of surgeons and found an improvement in HRV parameters during simulated surgery [45]. To our knowledge, this is the first study on an SMI for persons in leadership positions that investigated HRV as an outcome.

## 2. Materials and Methods

This study represented a prospective, single-center, two-arm open-label randomized controlled trial investigating the effects of an SMI on HRV measurements using linear mixed-effects regression models.

### 2.1. Participants and Procedures

The details of the recruitment procedure and intervention can be found in Limm et al. [43]. In short, all lower- and middle-level managers (N = 262) working in the international manufacturing plant of a large vehicle production company located in Germany were approached, each responsible for the management of about 50 workers. The strain of this management position is to achieve the goals set by higher management, such as an annual productivity increase of 10% at the time of conducting the study. The association of managers was positively disposed towards the study and supported recruitment by providing information about the study. In total, 174 out of the 262 eligible executives provided written informed consent and were randomized into either the intervention group or the control group (1:1), as shown in Figure 1. The intervention group received a 2-day training workshop on stress management at baseline and a half-day booster training after four and six months. The manual-based intervention included information about stress and its somatic impact, exercises to improve the awareness of and insight into stressful situations in the workplace, and tools and exercises for short-term (e.g., relaxation) and long-term (e.g., social networking and support) stress reduction that focused on individual resources. Measurements were taken at baseline and after 12 months.

Sociodemographic data including diagnoses, physical activity, and smoking were assessed using self-report questionnaires at baseline. Depressive and anxious symptoms were assessed using the Hospital Anxiety and Depression scale (HADS) [46]. The 14-item scale contains seven items each for depressive and anxious symptoms, which are scaled on a 4-point Likert scale and then summed, resulting in a range from 0 to 21, with higher values representing more symptoms. A result of 0–8 points is considered as no significant depressive symptoms. Effort-reward imbalance was measured with the effort-reward imbalance scale (ERI) [47]. This scale has several subscales. Here, we only discuss the effort subscale with six items and the reward subscale with eleven items. The items are summed and higher values represent more effort or reward.

BMI was measured at baseline. Blood pressure was measured twice at baseline, once in the sitting position and once after five minutes of rest, with a digital blood pressure instrument (Boso, Jungingen, Germany) and the average was calculated.

ECG was measured for 24 h at baseline and after 12 months using a Schiller Medilog AR© and HRV was calculated using Schiller Medilog software© with automated artifact correction. Only measurements with less than two artifacts per minute (2880 artifacts during 24 h) were entered into the analyses, regardless of the type of artifact (aberrant rhythm or movement artifact). Due to the very small number of women in the sample (N = 3) and the known sex differences in HRV [48], we restricted the analysis to men. Participants with cardiac diagnoses (e.g., atrial fibrillation) were also excluded. For the details of the data samples at each time point, see Figure 1.

The following parameters were extracted for analyses: heart rate (HR), SDNN (standard deviation of all RR intervals), SDANN (standard deviation of the average RR intervals of all 5-min segments of a measurement), RMSSD (square root of the squared mean of the sum of all differences of successive RR intervals). SDNN and SDANN especially represent the whole range of the ANS, including circadian influences, such as body temperature and the renin–angiotensin system, while RMSSD mirrors fast changes caused by the parasympathetic nervous system (Shaffer and Ginsberg, 2017). All parameters were calculated separately for the nighttime and 24 h periods. The nighttime periods were assessed by self-report measures.

The study was approved by the Ethics Committee of the University of Munich, Germany (no. 1439/05).

### 2.2. Statistical Methods

The intervention and control groups were compared using the Mann–Whitney U test as the majority of the variables was not normally distributed. Sociodemographic variables did not differ between the IG and CG.

A linear mixed-effects regression model with random intercepts was fitted using STATA© 15.1 (STATA Corp, College Station, TX, USA). The dependent variables were SDANN 24 h, log(SDANN nighttime), square root (SDNN 24 h), 1/square root (SDNN nighttime), log(RMSSD 24 h), log(RMSSD nighttime), square root (HR 24 h), and 1/square root (HR nighttime). Time was level 1 and the individual person was on level 2. A two-way interaction between the groups (control vs. intervention) and time (baseline vs. follow-up) was modeled using restricted maximum likelihood. The regression parameters were normalized by choosing the function that best approximated the Gaussian/normal distribution as marked. Age was included in the models as covariates in the fixed effect part. A significance level of *p* < 0.05 was regarded as significant. Marginal mean plots were calculated at average at fixed values for group and time interaction and averaging over the remaining covariate age.

## 3. Results

Table 1 shows the baseline characteristics for IG and CG. Participants were male, between 25 and 58 years old, and one in every four was a smoker. One in every three reported working more than 46 h per week. Sociodemographic variables did not differ between the intervention and control groups. The control group showed higher scores on the reward scale.

SDANN, SDNN, RMSSD, and HR are reported in Table 2. No differences were observed between IG and CG, nor were any differences between baseline and follow-up results for the whole study sample. A significant group × time interaction was found for nighttime SDANN (z = 2.04, *p* = 0.041) (Table 3). Further analyses showed that nighttime SDANN improved significantly in the intervention group compared to the control group (Figure 2). No significant changes were shown for SDANN (24 h), RMSSD (24 h and nighttime) or HR (24 h and nighttime).

## 4. Discussion

The stress intervention achieved its goal of improving a biological measure. Improvements in subjective stress reactivity have been shown before [43]. However, changes in the physiological levels were small. The only HRV parameter showing a significant increase in the intervention group was the SDANN.

SDANN is an estimate of the long-term components of HRV [49], which is highly correlated with the ultra-low frequency band of frequency domain HRV parameters. Next to ANS activity, circadian rhythms, such as body temperature and the renin–angiotensin system, contribute to this variability measure [18]. More importantly, SDANN has been shown to reflect mean HR and the coefficient of variation [50]. The latter is strongly shaped by the range of physical activities an individual conducts throughout the day. Therefore, it corresponds to functional capacity. Both the mean HR [51] and functional capacity [52] are strongly associated with mortality. Nighttime HR has even been shown to be superior to 24 h HR in predicting cardiovascular events in hypertensive patients [53]. SDANN is also strongly correlated with minimum HR [18]. As improvements in SDANN occurred in our sample during sleep, physical activity was probably not the cause. As explained above, the other main source of SDANN is the levels of HR. As mean HR did not change significantly in our sample, the change in SDANN might be caused by a reduced minimum HR during the night. HR most decreased during slow-wave sleep or deep sleep, accompanied by a reduction in blood pressure, sympathetic nervous activity, and cortisol release, thereby exerting important restorative functions [54]. Therefore, the improvement in nighttime SDANN might correspond to improved sleep quality. Improvements in sleep quality after stress management interventions have been previously shown, but only in subjective measures [55,56].

There were no changes in more specific autonomic markers, such as RMSSD. Research on changes in HRV parameters during primary prevention is scarce. Results from coronary artery disease populations have shown that it is possible to increase HRV, but the absolute numbers were small and therefore, study sizes need to contain several hundred participants [57]. There have also been results showing no changes at all, even with high treatment doses, in somatoform patients [58]. It is probable that an SMI can only change HRV when it triggers longstanding changes in attitude, i.e., learning how to recognize and cope with stress at an early stage. Behaviors that increase HRV include sport [59], nutrition [60], relaxation and breathing techniques [61], and biofeedback [62]. BMI, nutrition behavior, and physical activity did not change over time in the intervention group nor in the control group, while the attitude toward stressful situations, as measured by the stress reactivity score, changed significantly [43]. The latter might have contributed to the better sleep that was detected by the SDANN without inducing behavior changes that would have resulted in improved RMSSD. As reduced nighttime vagal activity is associated with adverse health outcomes [25,26], it would be desirable to establish an intervention that also improves vagal measures, such as RMSSD.

The population studied here was under high work strain with an effort score (ERI) that was markedly higher than reported in other European samples [63]. They showed unfavorable health behavior, with 33% being regular smokers and three quarters having a BMI that indicated adiposity, which is more than the 43% that has been reported for middle-aged men in the German population [64]. Still, blood pressure values were within the mean range of population values [65] and the mean resting HR and HRV parameters did not differ substantially from the 50th percentile of other German populations [66,67]. Our study population represented a target population of SMIs suffering from relevant stress, but without having already developed physiological or longer-lasting health consequences.

Some limitations applied to this study. First, the study sample might be biased by the selection of managers who were interested in the topic. Even though the study was supported by the manager’s association and thereby achieved a participation rate of 66%, one third of the managers could not be surveyed. It remains unclear whether they perceived themselves as having no stress in their workplace and were therefore not interested in the study or whether they were so burdened that they could not imagine sacrificing the required time for it. Further studies in this area could try to establish the reasons from at least some of the non-participants. Another limitation was the loss to follow-up, which was primarily due to the long follow-up duration of 12 months. Due to usual staff turnover, some managers left the company within the year and some others retired. To reduce this phenomenon, a follow-up measurement after only 3 or 6 months would be useful. Further, the study population primarily consisted of men (>95%) from a technical industry, working in lower- and middle-level management. Thus, the sample was not representative for employees nor for managers in general. Furthermore, changing environmental factors, such as periods with higher or lower workloads, were not assessed. Though 24 h measurements offered the advantage of also covering the nighttime, the disadvantage was the uncontrolled environment, which contributed to a high variance in measurement values.

Overall, the effects of this SMI on a biological level were minor. A broader approach is probably needed, covering not only individual but also organizational topics and providing regular refreshers. Few studies have applied a complex intervention addressing both levels, for example, the SEEGEN trial [68]. Programs such as this should form part of an integrated intervention approach [69], including occupational health physicians [70]. The authors of a previous Cochrane meta-analysis among health care workers on interventions addressing either the organizational or individual level concluded that for individual approaches, cognitive behavioral training (CBT) as well as mental and physical relaxation led to a moderate stress reduction. Interventions on the organizational level, such as changes in working conditions, organizing support, changing care or increasing communication skills, were not more effective than control groups [71].

An ongoing study of N = 200 leaders in small and medium-sized enterprises is replicating the present SMI by investigating psychophysiological parameters, such as HRV and cortisol, and also cost-effectiveness [72]. This study was funded by the German Federal Ministry of Education and Research (Bundesministerium für Bildung und Forschung, BMBF) under the funding references 01EL2003A, 01EL2003B, and 01EL2003C (German Clinical Trial Register: DRKS00023457).

## 5. Conclusions

The male lower- and middle-level managers in this study benefited from the SMI with a small but significant improvement in nocturnal SDANN 12 months after the intervention in addition to reduced subjective stress reactivity. As SDANN is associated with minimum heart rate, which occurs during deep sleep, this improvement could correlate with improved sleep quality, which would in turn be consistent with reduced subjective stress reactivity. To enhance these effects, training could be provided to at-risk groups that are selected due to either high subjective stress reports or reduced physiological measures, such as HRV, or a combination of both. In addition, physiological measures could be used as a part of the intervention via feedback on measurement results [66]. The limitations of this study include selection bias, as well as other sources of influence due to the long follow-up period of 12 months. Future studies should further explore the dose and the kind of input necessary for physiological changes, as well as the necessity for repeated interventions to maintain the effect.

## Figures and Tables

**Figure 1 ijerph-19-03841-f001:**
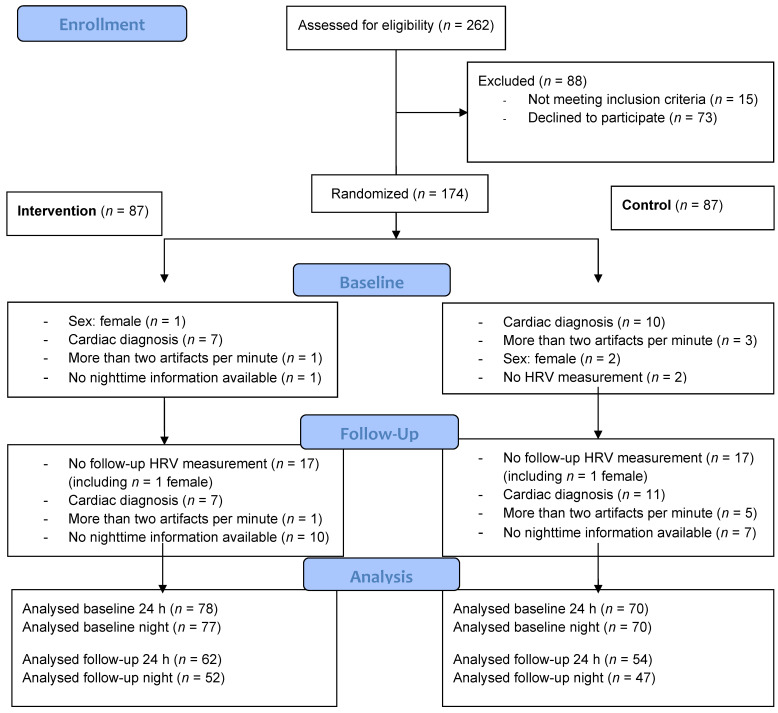
The recruitment flowchart.

**Figure 2 ijerph-19-03841-f002:**
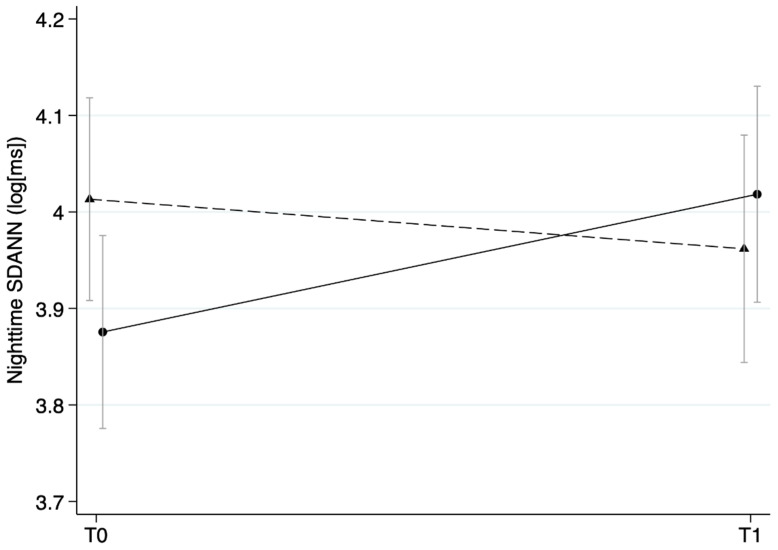
The predicted scores of log(SDANN) (ms) at baseline (Pre) and follow-up (Post) for the intervention and control groups (mean ± SD).

**Table 1 ijerph-19-03841-t001:** The characteristics of the study population.

	Intervention Group N = 87			Control Group N = 87			
Variable	25th Quartile (N)	Median (%)	75th Quartile (N)	25th Quartile (N)	Median (%)	75th Quartile (N)	*p*-Value
Age (years)	35	41	47	35	41	48	0.922
Sex (female)	1	1		2	2		1.000
Smoker	22	25		29	33		0.318
BMI (kg/m^2^)	25.3	27.8	30.7	25.3	27.5	29.5	0.217
Resting systolic blood pressure (mmHg)	124	134	141	125	133	142	0.945
Resting diastolic blood pressure (mmHg)	81	90	94	82	88	94	0.789
Resting heart rate (bpm)	70.2	75.1	82.2	70.9	77.1	82.5	0.823
Weekly working hours:							0.743
<40	17	20		20	23		
41–45	44	51		40	46		
46–50	21	24		19	22		
>50	5	6		8	9		
Sick leave during the last 12 months: More than 10 days	11	13		11	13		1.000
HADS anxious symptoms	4	5	8	3	6	9	0.614
HADS depressive symptoms	2	4	7	2	4	6	0.189
ERI effort	14	16	19	14	16	18	0.757
ERI reward	38	43	49	43	47	51	0.003

The *p*-values were calculated using the Mann–Whitney U test for the metric and chi-squared tests for dichotomous variables. Abbreviations: HADS, Hospital Anxiety and Depression Scale; ERI, effort-reward imbalance scale.

**Table 2 ijerph-19-03841-t002:** HRV parameters at baseline and follow-up.

Parameter	Group	Baseline			Follow-Up			Baseline			Follow-Up		
		Night			Night			24 h			24 h		
		N	Mean	SD	N	Mean	SD	N	Mean	SD	N	Mean	SD
SDANN	CG	70	61.4	32.1	47	56.8	23.4	70	126.0	36.8	54	122.0	31.7
SDANN	IG	77	53.5	25.6	52	62.5	29.4	78	131.0	34.5	62	125.0	35.5
SDNN	CG	70	102.0	34.4	47	101.0	33.2	70	142.0	34.3	54	143.0	35.0
SDNN	IG	77	99.4	29.7	52	107.0	30.8	78	150.0	36.4	62	145.0	35.3
RMSSD	CG	70	36.4	16.5	47	36.9	15.8	70	26.9	9.4	54	28.1	10.1
RMSSD	IG	77	38.4	17.2	52	39.9	17.9	78	29.0	10.4	62	29.7	11.5
HR	CG	70	64.5	8.5	47	63.4	10.1	70	77.5	8.5	54	75.9	9.7
HR	IG	77	63.0	8,2	52	62.2	7.2	78	76.4	8.8	62	76.0	9.2

Abbreviations: 24 h, 24 hours; CG, control group; HR, heart rate; IG, intervention group; RMSSD, root mean square of successive differences; SDANN, standard deviation of the average of normal-to-normal intervals; SDNN, standard deviation of normal-to-normal intervals.

**Table 3 ijerph-19-03841-t003:** The results of linear mixed-effects regression models for HRV parameters.

Dependent Variables			Group ×Time		Group		Time	
		N	z	*p*-Value	z	*p*-Value	z	*p*-Value
SDANN	Nighttime	149	2.04	0.041 *	−1.86	0.063	−0.75	0.455
SDANN	24 h	150	−0.13	0.898	0.88	0.378	−1.38	0.166
SDNN	Nighttime	149	1.23	0.218	−0.49	0.623	−0.38	0.701
SDNN	24 h	150	−0.48	0.631	1.37	0.170	−0.83	0.405
RMSSD	Nighttime	149	0.22	0.825	0.98	0.329	−0.38	0.703
RMSSD	24 h	150	−0.53	0.594	1.41	0.157	0.69	0.487
HR	Nighttime	149	−0.25	0.799	1.13	0.259	0.26	0.796
HR	24 h	150	0.83	0.409	−0.75	0.453	−0.82	0.412

The *p*-values were calculated from the linear mixed-effects model with random intercepts with age as a covariate. * *p* < 0.05. Abbreviations: 24 h, 24 hours; HR, heart rate; RMSSD, root mean square successive differences; SDANN, standard deviation of the average of normal-to-normal intervals; SDNN, standard deviation of normal-to-normal intervals.

## Data Availability

The data that support the findings of this study are available from the corresponding author upon reasonable request.

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
