# Peer review of "Stress Management Intervention for Leaders Increases Nighttime SDANN: Results from a Randomized Controlled Trial"

_ijerph, 2022, doi:10.3390/ijerph19073841_

Round 1

Reviewer 1 Report

The paper provides a good approach abour randomized controlled trial to improve stress management controlling HRV. The work is well structured and organized. 
I suggest to add in the introduction more references about prespectives (artificial intelligence in telemedicine platform monitoring heart parameters), such as all the listed one:

https://doi.org/10.3390/s21186296
https://doi.org/10.3390/ijerph19042417
DOI: 10.23919/AEIT.2018.8577362
https://doi.org/10.3390/s21113719
DOI: 10.1109/MetroInd4.0IoT48571.2020.9138258
https://doi.org/10.3390/s22010034

Conclusions should be improved by starting to the discussion about recruitment (Fig. 1) and arriving to enhance the obtained results
A minor revision is required.

Author Response

Thank you for your valuable comments.

R:"I suggest to add in the introduction more references about prespectives (artificial intelligence in telemedicine platform monitoring heart parameters), such as all the listed one:"

We gladly introduced a sentence highlighting future possibilities with DL and AI in the introduction including the proposed citations:

... With advances in digital health technologies, techniques using Deep Learning, artificial intelligence and neural networks are emerging [35–38], some of which can approximate heart rate without a device worn directly on the skin [39,40]. This rapidly developing area holds the potential for widespread application, especially for the field of preventive medicine at the workplace.  ...

R: Conclusions should be improved by starting to the discussion about recruitment (Fig. 1) and arriving to enhance the obtained results

Thank you for this valuable hint. We extended the conclusion starting from the recruitment:

Conclusion:

The male lower and mid-level managers benefited from the SMI with a small but significant improvement in nocturnal SDANN twelve months later, in addition to reduced subjective stress reactivity. Because SDANN is associated with minimal heart rate and the latter occurs during deep sleep, this improvement could correlate with improved sleep quality, which in turn would be consistent with reduced subjective stress reactivity. To enhance these effects, training could be provided to at-risk groups selected by either high subjective stress reports or reduced physiological measures such as HRV, or a combination of both. In addition, physiological measures could be used as part of the intervention via feedback on measurement results [66]. Limitations of the study include selection bias as well as other sources of influence due to the long follow-up period of twelve months. Future studies should explore more deeply the dose and the kind of input necessary for physiological changes as well as the necessity for repeated interventions to hold the effect.

The manuscript was spell-checked.  

Reviewer 2 Report

In the material and methods it is necessary to highlight the study model, the selection of the sample, a small paragraph listing the study model and statistical methods used right at the beginning, as well as the code of the scientific committee that authorized the study. Check when you use 24h and then use 24 hours, keep the same nomenclature in the text. The tables are very simple highlight the most important columns. You should highlight the limitations that you have encountered in the development of your work.
You should review those citations that are not referents of your contextualization and are more than 5 years old, this would increase the quality of your work.              

Author Response

Point 1: In the material and methods it is necessary to highlight the study model, the selection of the sample, a small paragraph listing the study model and statistical methods used right at the beginning, as well as the code of the scientific committee that authorized the study.

Response 1: We thank the reviewer for the valuable comments. We introduced a sentence summarizing study characteristics in materials and methods and extended description of the sample selection/recruitment and added the code of the ethics committee:

Materials and Methods

This study represents a prospective, single-center, two-arm open-label randomized controlled trial investigating the effects of an SMI on measures of HRV using linear mixed-effects regression models.  

2.1. Participants and procedures:

Details of the recruitment procedure and the intervention can be found in Limm et al. [43]. In short, all lower and middle-level managers (N=262) of an international manufacturing plant of a large vehicle production company located in Germany, each responsible for the management of about 50 workers, were approached. The strain of this management position is to achieve the goals set by higher management, such as an annual productivity increase of 10% at the time of conducting the study in contact with the executing employees. The association of managers was positively disposed towards the study and supported the recruitment by providing information about the study…

Institutional Review Board Statement: The study was conducted in accordance with the Declaration of Helsinki, and approved by the Ethics Committee of THE UNIVERSITY OF MUNICH; GERMANY: No. 1439/05.

Point 2: Check when you use 24h and then use 24 hours, keep the same nomenclature in the text.

Response 2: Thank you for this hint, we unified the use of 24h.

Point 3: The tables are very simple highlight the most important columns.

Response 3: Our attempts to highlight columns were graphically unconvincing, but we rearranged the tables starting with the most important variables SDANN as a kind of highlighting. Further, we added some explanatory sentences in the results section.

Point 4: You should highlight the limitations that you have encountered in the development of your work.

Thank you for this important comment. We added limitations, especially in recruitment and follow-up, to the discussion:

Some limitations apply. First, the study sample might be biased by the selection of managers interested in the topic. Even though the study was supported by the manager’s association and thereby achieved a participation rate of 66%, one-third of the managers could not be surveyed. It remains unclear whether they perceived no stress at their workplaces and were therefore not interested in the study, or whether they were so burdened that they could not imagine sacrificing time for it. Further studies in this area could try to find out the reasons from at least some of the non-participants. Another limitation is the loss-to-follow-up which is primarily due to the long follow-up duration of 12 months.  Due to usual staff turnover, some managers left the company after the year, others retired. To reduce this phenomenon, a follow-up measurement after only 3 or 6 months would be useful.

Point 5: You should review those citations that are not referents of your contextualization and are more than 5 years old, this would increase the quality of your work.        

Response 5: Thanks for the hint with the citation, we reviewed them and updated a substantial number. Unfortunately, changed citations are not visible when tracking in word.

Reviewer 3 Report

The issues discussed by the authors are interesting and constantly relevant. The presented research model allows the reader to recognize the authors' research intention relatively easily. Selection of people for the research, the applied measurement methods, descriptive statistics and statistical inference are completely correct.
The obtained results, although they do not show a significant discrepancy in the physiological responses resulting from the intervention (and only at night), were insufficiently used in the conclusions of the study. This is the fragment of the article that should be supplemented (from line 264). The cognitive and application possibility of using interventions should be described more broadly, e.g. in the form of specially prepared training for selected risk groups, and then the limitations of research and their further directions should be indicated.

Author Response

Point 1:The issues discussed by the authors are interesting and constantly relevant. The presented research model allows the reader to recognize the authors' research intention relatively easily. Selection of people for the research, the applied measurement methods, descriptive statistics and statistical inference are completely correct.
The obtained results, although they do not show a significant discrepancy in the physiological responses resulting from the intervention (and only at night), were insufficiently used in the conclusions of the study. This is the fragment of the article that should be supplemented (from line 264). The cognitive and application possibility of using interventions should be described more broadly, e.g. in the form of specially prepared training for selected risk groups, and then the limitations of research and their further directions should be indicated.

Response 1: Thanks for this important comment. We extended the conclusions including application possibilities:

The male lower and mid-level managers benefited from the SMI with a small but significant improvement in nocturnal SDANN twelve months later, in addition to reduced subjective stress reactivity. Because SDANN is associated with minimal heart rate and the latter occurs during deep sleep, this improvement could correlate with improved sleep quality, which in turn would be consistent with reduced subjective stress reactivity. To enhance these effects, training could be provided to at-risk groups selected by either high subjective stress reports or reduced physiological measures such as HRV, or a combination of both. In addition, physiological measures could be used as part of the intervention via feedback on measurement results [66]. Limitations of the study include selection bias as well as other sources of influence due to the long follow-up period of twelve months. Future studies should explore more deeply the dose and the kind of input necessary for physiological changes as well as the necessity for repeated interventions to hold the effect.

The manuscript was spell-checked.